# Preparation and Properties of (Cu, Ni) Co-Doped ZnO Nanoparticle-Reinforced Cu-Ni Nanocomposite Coatings

**DOI:** 10.3390/ma16072746

**Published:** 2023-03-29

**Authors:** Haifeng Tan, Chunlin He, Jie Yang, Haixuan Sunyu, Yunhe Ling, Jinlin Zhang, Guihong Song

**Affiliations:** 1Liaoning Provincial Key Laboratory of Advanced Materials, Shenyang University, Shenyang 110044, China; 2AVIC Electromechanical (Shenyang) Sanyo Refrigeration Equipment Co., Ltd., Shenyang 110020, China; 3School of Material Science and Technology, Shenyang University of Technology, Shenyang 110870, China

**Keywords:** (Cu, Ni) co-doped ZnO, Cu-Ni nanocomposite coating, microhardness, corrosion resistance, photocatalytic properties

## Abstract

Here, 2% Cu + 2% Ni co-doped ZnO nanoparticles were synthesized using the hydrothermal method and were used as particle reinforcements of Cu-Ni nanocomposite coatings prepared by electroplating technology. The effects of the added (Cu, Ni) co-doped ZnO nanoparticles (2–8 g/L) on the phase structure, surface morphology, thickness, microhardness, corrosion resistance, and photocatalytic properties of the coatings were investigated. The nanocomposite coatings have obvious diffraction peaks on the crystal planes of (111), (200), and (220), showing a wurtzite structure. The surface of the nanocomposite coatings is cauliflower-like, and becomes smoother and denser with the increase in the addition of nanoparticles. The grain size, thickness, microhardness, corrosion resistance, and photocatalytic properties of the nanocomposite coating reach a peak value when the added (Cu, Ni) co-doped ZnO nanoparticles are 6 g/L. At this concentration, the mean crystallite size of the coating reaches a minimum of 15.31 nm, and the deposition efficiency of the coating is the highest. The (Cu, Ni) co-doped ZnO nanoparticle reinforcement makes the microhardness reach up to 658 HV. The addition of nanoparticles significantly improves the corrosion resistance and photocatalytic properties of nanocomposite coatings. The minimum corrosion current density is 2.36 × 10^−6^ A/cm^2^, the maximum corrosion potential is −0.301 V, and the highest decolorization rate of Rhodamine B is 28.73% after UV irradiation for 5 h.

## 1. Introduction

Marine corrosion is an increasingly serious global problem, which not only causes enormous economic losses, but also brings serious personnel safety and environmental pollution problems. It is a key problem that needs to be solved for economic development. It is estimated that the global annual corrosion costs are USD 4 trillion [1], of which about 20% is caused by some form of microbial activity [2]. At present, active metals such as Al [3], Mg [4], Fe [5], and Cu [6], as well as their alloys, are facing significant challenges in complex marine environments.

The preparation of coatings is a simple, economical, and feasible metal surface protection technology that can not only improve the seawater corrosion resistance and marine microbial corrosion resistance of structural parts, but also strengthen the friction and wear resistance. Cu-Ni alloy parts and Cu-Ni alloy coatings have excellent anti-fouling, anti-corrosion, durability, and good strength in seawater [7]. Therefore, they have been widely used in the field of marine engineering, such as boiler components, heat exchanger tubes, boat hulls, seawater condensers, valve bodies, oil platforms, and other ship hardware [8]. Numerous studies have shown that Cu-Ni alloy coatings prepared by electroplating technology have a high strength and excellent corrosion resistance in many harsh environments, such as seawater, oxidizing and reducing gas environments, and alkaline and acidic media [9,10,11]. However, Cu-Ni alloy coatings still cannot meet the rigorous service requirements of the complex and changeable marine environments.

The addition of nanoparticles, such as Al_2_O_3_ [12], ZrO_2_ [13], TiN [14], MMT [9], Y_2_O_3_ [15], and Gr [16], can often further improve the strength, hardness, friction and wear resistance, and corrosion resistance of Cu-Ni coatings, but it is difficult to improve the microbial corrosion resistance. As a wide band gap semiconductor material, ZnO has been widely studied due to its potential applications in photocatalysist [17], antiseptic [18], and semiconductor devices such as UV photodetectors, storage, solar cells, gas sensors, displays, light emitting diodes (LED), and piezoelectric devices [19,20,21,22]. However, ZnO not only has a wide band gap (3.37 eV), but also a high exciton binding energy (60 meV), which hinders its photocatalysis under solar energy [23]. By doping transition metal ions in ZnO, its electronic structure can be changed, thereby changing its electrical and optical properties [24]. The common dopants in ZnO-based systems are mainly transition metals, including Mn, Fe, Co, Ni, Cu, etc. [17,25,26,27].

The introduction of transition metal ions in ZnO, such as Fe, Co, and Ni, has a significant effect on its photoelectric properties, especially UV absorption and electrical conductivity [28,29]. Cu has a similar electron shell structure to Zn [30], which can affect the band gap of ZnO [31]. As the atomic radius of Cu and Ni is smaller than that of Zn, they can be used to replace the ZnO lattice to tune its photocatalytic activity [25,32]. Some related research results have shown that Cu and Ni co-doped ZnO can significantly affect UV absorption and luminescence properties, thereby stimulating the photocatalytic activity [25,33,34,35]. A series of ZnO nanocomposites doped with different amounts of Cu and Ni were prepared using the hydrothermal method in our previous work [36]. The results of this work and subsequent work showed that the photocatalytic activity of the 2% Cu + 2% Ni co-doped ZnO nanocomposite was significantly better than pure ZnO, single-doped ZnO, and other amounts of (Cu, Ni) co-doped ZnO. The degradation rate of rhodamine B (RhB) could reach 92.45% after irradiating with a 10 W ultraviolet lamp for 5 h. Therefore, the doped ZnO nanoparticles will provide a possibility for the improvement of the corrosion resistance and photocatalytic sterilization performance of Cu-Ni coatings.

In this paper, Cu and Ni were doped into ZnO nanoparticles by the hydrothermal method to improve the photocatalytic performance. The doped ZnO nanoparticles were effectively combined with Cu-Ni coatings by electroplating technology. The effects of process parameters such as temperature, current density, deposition time, and pH value on the performance of nanocomposite coatings have been studied in an unpublished work. This work used a set of better electroplating process parameters and mainly studied the effects of (Cu, Ni)-ZnO nanoparticle addition in the electrolytic solution on the microstructure, mechanical properties, corrosion resistance, and photocatalytic degradation properties of Cu-Ni nanocomposite coatings. The purpose is to attempt to introduce nanoparticles with photocatalytic properties into nanocomposite coatings to expand the application of active metals under marine environments. Furthermore, nanoparticles deposited on the surface of metal alloy by electroplating technology can effectively improve the utilization rate and solve the problem that powder is difficult to recover.

## 2. Materials and Methods

### 2.1. Synthesis of (Cu, Ni)-ZnO Nanoparticles

Cu and Ni co-doped ZnO nanoparticles were prepared by a simple hydrothermal method reported in a previous work [36]. Zinc nitrate hexahydrate [Zn(NO_3_)_2_ ∙ 6H_2_O], copper nitrate trihydrate [Cu(SO_4_)_2_ ∙ 3H_2_O], nickel nitrate hexahydrate [Ni(NO_3_)_2_ ∙ 6H_2_O], and hexamethylenetetramine [C_6_H_12_N_4_ (HMT)] were used as precursor solutions, and trina citrate dihydrate [C_6_H_5_Na_3_O_7_ ∙ 2H_2_O] was used as a surfactant. The molar ratio of Zn^2+^ to HMT in the precursor mixed solution was 1: 1. After 5 min of magnetic stirring and 5 min of ultrasonic dispersion, the mixture was transferred to a reaction kettle and kept at 90 °C for 4 h. The obtained products were filtered and repeatedly washed several times with deionized water until no bubbles were generated in the filter bottle. Then, they were washed three times with anhydrous ethanol, and dried in a drying oven at 80 °C for 6 h. The dried products were calcined in a muffle furnace at 500 °C for 2 h, and Cu, Ni co-doped ZnO nanoparticles were obtained after grinding. (Cu, Ni)-ZnO nanoparticles with different Cu, Ni, and Zn molar ratios can be prepared by changing the amount of Cu(SO_4_)_2_ ∙ 6H_2_O and Ni(NO_3_)_2_ ∙ 6H_2_O. In this experiment, 2 at% Cu + 2 at% Ni co-doped ZnO nanoparticles were prepared.

### 2.2. Preparation of Cu-Ni-ZnO Nanocomposite Coatings

In this experiment, a 2024 aluminum alloy plate of 25 mm × 25 mm × 2 mm was used as the cathode of electroplating, and 70–30 Cu-Ni alloy of 20 mm × 30 mm × 3 mm was selected as the anode. The aluminum alloy substrate was first mechanically polished, then immersed into a solution of 0.2 g/L NaOH, 20 g/L Na_3_PO_4_, and 20 g/L Na_2_CO_3_, and degreased at 55 °C for 3–5 min. In order to completely remove the oxides on the surface of the substrate, the aluminum alloy cathode after alkali washing needed to be immersed into a solution of 7 mol/L HNO_3_ and 5.5 mol/L HF solution at 55 °C for 5–7 s. Nanocomposite electroplating uses a simple DC stabilized power supply, and the composition and process parameters of the electroplating solution are shown in Table 1. The addition of 2% Cu + 2% Ni co-doped ZnO nanoparticles ranged from 2 to 8 g/L. In order to prevent the agglomeration of nanoparticles, the magnetic stirrer was used to strongly stir at 400 rpm for 2 h before electroplating to ensure the uniform dispersion of nanoparticles in the electrolyte.

### 2.3. Characterization Techniques

The crystal structure of nanoparticles and nanocomposite coatings was analyzed using an X-ray diffractometer (Tongda, TD-3500) using Cu Kα radiation (*λ* = 1.5406 Å) at 35 kV and 50 mA. Diffraction Angle 2*θ* ranged from 20–80° in the scan speed of 12 °/min. The surface morphology, chemical composition, and thickness of the nanocomposite coatings were characterized by scanning electron microscopy (SEM, Hitachi S-4800). The chemical compositions were investigated by energy disperse spectroscopy (EDS, Oxford, UK). The hardness of the nanocomposite coatings was tested using 402MVD digital Vickers hardness tester.

The effects of the added (Cu, Ni) co-doped ZnO nanoparticles on the corrosion behavior of nanocomposite coatings were studied in a three-electrode cell with a CHI 604E device. The prepared nanocomposite coatings were used as the working electrode, a saturated calomel electrode was used as the reference electrode, and a graphite electrode was used as the counter electrode. Corrosion resistance testing was performed in a 3.5% NaCl solution at room temperature. Electrochemical impedance spectroscopy (EIS) measurement was conducted at *E*_ocp_ in 10^5^ Hz-10^−2^ Hz with an a.c. excitation potential amplitude of 10 mV. Potentiodynamic polarization curves were obtained by changing the electrode potential automatically from *E*_ocp_ −500 mV to *E*_ocp_ +800 mV at a potential scan rate of 0.166 mVs^−1^. The fitting results of the equivalent circuit of Cu-Ni-ZnO nanocomposite coatings were obtained by a conventional fitting method.

The photocatalytic performance of the prepared nanocomposite coatings on RhB solution under ultraviolet light irradiation was investigated. The prepared nanocomposite coatings were immersed in 100 mL of 8 mg/L RhB solution and were fully stirred in the dark for 60 min to achieve adsorption equilibrium. Under room temperature, a 10 W UV lamp was used to irradiate, and the degradation process was detected by measuring the absorbance with a UV-VIS spectrophotometer. The decolorization rate *η* of the nanocomposite coatings in the RhB solution was calculated as follows: (1)$$\eta =\left(C_{0}-C_{t}\right)/C_{0}\times 100\%=\left(A_{0}-A_{t}\right)/A_{0}\times 100\%$$
where *C_0_* is the initial concentration of RhB solution and *C_t_* is the concentration at a certain time of photocatalysis. Further conversion, *A_0_* is the initial absorbance of RhB solution, and *A_t_* is the absorbance at a certain time of photocatalysis. 

## 3. Results

### 3.1. Phase Structure

Figure 1 shows the XRD analysis results of the (Cu, Ni)-ZnO nanopowder and Cu-Ni-ZnO nanocomposite coatings. The crystal structure of the 2% Cu + 2% Ni co-doped ZnO nanopowder was very close to the standard pure ZnO hexagonal wurtzite (PDF No. 76-0704) [36]. Cu and Ni co-doping made the diffraction peak of ZnO shift to a large angle direction, which may be due to the slightly smaller ionic radii of Cu^2+^ (0.072 nm) and Ni^2+^ (0.069 nm) than that of Zn^2+^ (0.074 nm) [37,38]. Therefore, when Cu and Ni were doped into ZnO, lattice collapse was caused, resulting in a shift in the ZnO diffraction peak. The nanocomposite coatings exhibited Cu(Ni) (111), (200), and (220) reflections at around 43.3°, 50.4°, and 74.1°, respectively, and showed a dominant orientation of the (111) reflection, regardless of the (Cu, Ni)-ZnO addition. Compared with Cu-Ni coating, the diffraction angle of the Cu-Ni-ZnO nanocomposite coatings gradually shifted to a larger angle direction with the increase in (Cu, Ni)-ZnO additions.

The average crystallite size was estimated using the Debye–Scherrer formula [39]: (2)$$D=\frac{K\lambda }{\beta \cos\theta }$$
where *D* is the mean crystalline dimension normal to diffracting planes, the Scherrer constant *K* is 0.91, X-ray wavelength *λ* is 0.15406 nm, *β* in radian is the peak width at half-maximum height, and *θ* is the Bragg’s angle. Figure 2 presents the calculated mean crystallite sizes of the Cu-Ni solid solution crystallites based on the diffraction peak of the (111) crystal plane as a function of the (Cu, Ni)-ZnO additions. It can be seen from the figure that the mean crystallite sizes of all of the electrodeposited coatings decreased from 17.3 nm to 15.5 nm when the (Cu, Ni)-ZnO additions increased from 0 to 8 g/L, showing that the additional nanoparticles were beneficial to grain refinement [40,41,42]. However, the refinement effect was not obvious, perhaps associated with the relatively lower content of nanoparticles in the coatings. The refinement effect resulting from (Cu, Ni)-ZnO nanoparticles included the following: (i) (Cu, Ni)-ZnO nanoparticles were located at the grain boundaries of the Cu-Ni solid solution, hindering grain growth, and (ii) some (Cu, Ni)-ZnO nanoparticles could act as nucleation centers of Cu-Ni crystals. Therefore, adding (Cu, Ni)-ZnO nanoparticles could effectively reduce the mean crystallite size of the nanocomposite coatings, and the structure was more detailed and uniform. There was no diffraction peak of ZnO in the XRD patterns of the Cu-Ni-ZnO nanocomposite coatings, because the content of (Cu, Ni)-ZnO nanoparticles in the coatings was too small [13,15]. 

### 3.2. Surface Morphology

Figure 3 shows the surface morphology of the (Cu, Ni)-ZnO nanoparticles and Cu-Ni-ZnO nanocomposite coatings with different additions of Cu and Ni co-doped ZnO nanoparticles. The (Cu, Ni)-ZnO nanoparticles were uniform in size, with an average size of 60 nm, and had good dispersion, as shown in Figure 3a. The surface of the Cu-Ni nanocomposite coating without adding (Cu, Ni)-ZnO nanoparticle was cauliflower-like, with large cell-like particles. There were a lot of gaps and holes, resulting in significantly poor compactness. With the increase in (Cu, Ni)-ZnO additions, the cell-like particles on the surface of Cu-Ni-ZnO nanocomposite coatings became more uniform and compact. The increase in (Cu, Ni)-ZnO additions in the electroplating solution was conducive to the deposition of more nanoparticles onto the cathode surface. Relevant research results show that embedded reinforcing particles can fill existing defects (such as microcracks, pores, etc.) and lead to dense and defect-less deposits [15,43,44]. This indicates that the addition of (Cu, Ni)-ZnO nanoparticles is beneficial to the refinement and densification of nanocomposite coatings.

Figure 4 shows the effect of nanoparticle addition on the content of the main elements of Cu, Ni, and Zn in Cu-Ni-ZnO nanocomposite coatings. When the addition of (Cu, Ni)-ZnO nanoparticles increased from 0 g/L to 6 g/L, the content of Zn in Cu-Ni-ZnO nanocomposite coatings increased continuously, and the content of Cu and Ni decreased gradually. The increase in the Zn element in the nanocomposite coatings represented the increase in (Cu, Ni)-ZnO nanoparticle content, which is beneficial to the grain refinement of Cu-Ni-ZnO nanocomposite coatings. When the addition of (Cu, Ni)-ZnO increased to 8 g/L, the content of Zn in Cu-Ni-ZnO nanocomposite coating decreased. With the continuous increase in the added nanoparticles, the nanoparticles in the plating solution reached a saturated state. The (Cu, Ni)-ZnO adsorbed on the surface of the coating was easy to wash away using the plating solution under the action of mechanical agitation before co-deposition with metal ions. At the same time, excessive (Cu, Ni)-ZnO nanoparticles were prone to agglomeration in the plating solution, and were not easy to move and deposit on the surface of the cathode, which reduced the concentration of nanoparticles in the coatings [45]. Therefore, reducing the (Cu, Ni)-ZnO content in the nanocomposite coatings resulted in an increase in the grain size of the nanocomposite coatings, which was consistent with the XRD results.

### 3.3. Microhardness

Figure 5 shows the relationship between the addition of (Cu, Ni)-ZnO and the thickness of the Cu-Ni-ZnO nanocomposite coatings after 45 min of deposition. With the increase in (Cu, Ni)-ZnO additions, the thickness in nanocomposite coatings showed a trend of increasing first and then decreasing. The thickness of Cu-Ni-ZnO nanocomposite coatings reached the peak value of 21.7 ± 0.4 μm when the addition in the plating solution was 6 g/L. This was due to the increase in (Cu, Ni)-ZnO additions, which improved the contact probability between the nanoparticles and the cathode matrix. More (Cu, Ni)-ZnO nanoparticles were adsorbed, which increased the number of catalytic active sites on the surface of the coatings and further improved the reduction rate of Cu and Ni. Therefore, adding more (Cu, Ni)-ZnO nanoparticles was beneficial for increasing the electroplating rate and it ultimately obtained thicker coatings. However, as the addition of (Cu, Ni)-ZnO continued to increase, the nanoparticles on the surface of the nanocomposite coatings tended to be saturated, which hindered the diffusion of Cu^2+^ and Ni^2+^, and reduced the electroplating speed and the coating thickness.

The microhardness of Cu-Ni-ZnO nanocomposite coatings with different additions of (Cu, Ni)-ZnO nanoparticles is shown in Figure 6. The substrate material of this experiment was the 2024 Al alloy with a microhardness of 108 HV. Compared with the substrate and Cu-Ni alloy coating, the microhardness of the Cu-Ni-ZnO nanocomposite coatings increased obviously. It has been reported that the strengthening mechanisms of particle reinforced metal matrix composites mainly include grain refinement, Orowan looping, load transfer, and the coefficient of the thermal expansion mismatch [46,47,48,49]. The added (Cu, Ni)-ZnO nanoparticles were uniformly dispersed in the nanocomposite coatings, and the Orowan strengthening mechanism played a significant role because the size of the particle reinforcement was less than 100 nm. The (Cu, Ni)-ZnO nanoparticles were subject to the load and hindered the movement of dislocations when the hardness tester was pressed into the composite coatings [50]. In contrast, indentation tips could easily penetrate deeper into the Al alloy substrate and pure Cu-Ni alloy coating. In addition, the (Cu, Ni)-ZnO nanoparticles uniformly distributed in the Cu-Ni coatings inhibited the plastic deformation of the nanocomposite coatings under load through grain refinement and dispersion strengthening.

With the increase in (Cu, Ni)-ZnO additions, the hardness of Cu-Ni-ZnO nanocomposite coatings increased continuously and reached the maximum value of 658 HV at 6 g/L. The more continuous the increase in (Cu, Ni)-ZnO content in the coatings, the better the strengthening effect, and thus the higher the hardness of the coatings. The results in Figure 4 also show that when the addition of (Cu, Ni)-ZnO in the electroplating solution was 6 g/L, the concentration of nanoparticles in the coatings reached the peak value, which led to the maximum hardness of the nanocomposite coatings. However, as the addition of (Cu, Ni)-ZnO continued to increase, the excessive nanoparticles in the bath tended to agglomerate, and the concentration of nanoparticles in the coatings decreased, resulting in a decrease in the microhardness of Cu-Ni-ZnO nanocomposite coatings.

### 3.4. Corrosion Resistance

Figure 7 shows the polarization curves of Cu-Ni-ZnO nanocoatings with different (Cu, Ni)-ZnO additions. It can be seen that, compared with the Cu-Ni alloy coating, the corrosion potential of the Cu-Ni-ZnO nanocomposite coatings was rather nobler (more positive), and the corrosion current density was lower. After Tafel fitting, the corrosion potential and corrosion current density of the Cu-Ni-ZnO nanocomposite coatings with different (Cu, Ni)-ZnO additions are shown in Table 2. 

According to the fitting results, the corrosion current density of Cu-Ni-ZnO nanocomposite coatings with the addition of 6 g/L (Cu, Ni)-ZnO nanoparticles was the lowest, 2.36 × 10^−6^ A ∙ cm^−2^, indicating that the corrosion rate of the coating was the slowest at the moment. The corrosion current density of the Al alloy substrate was also very low, which may be due to the formation of a dense alumina layer on its surface, slowing down the corrosion rate. With the increase in (Cu, Ni)-ZnO additions, the corrosion potential of Cu-Ni-ZnO nanocomposite coatings shifted positively and then negatively. The corrosion potential reached the noblest *E_corr_* of −0.301 V when the addition of (Cu, Ni)-ZnO in the plating solution increased to 6 g/L. Nevertheless, further increasing the (Cu, Ni)-ZnO additions, the corrosion current density increased sharply, and the corrosion potential shifted negatively.

Nyquist plots of Cu-Ni-ZnO nanocomposite coatings with different additions of (Cu, Ni)-ZnO nanoparticles are shown in Figure 8. It can be seen that the radius of the capacitive arc of the Cu-Ni-ZnO coatings increased continuously with the increase in (Cu, Ni)-ZnO nanoparticles additions. The corrosion resistance of Cu-Ni-ZnO nanocomposite coatings was the best at 6 g/L. In order to further study the corrosion performance of the Cu-Ni-ZnO nanocomposite coatings, the equivalent electrical circuit (EEC) for fitting the EIS of the nanocomposite coatings is shown in Figure 9, and the fitted corrosion parameters are listed in Table 3. 

The resistance *R_s_* is known as the solution resistance, *R_pore_* represents the micropore resistance of the coating surface, and the charge transfer resistance of the substrate is represented by *R_ct_*. *Q_1_* represents the coating capacitance and *Q_2_* is the electric double layer capacitance. When the (Cu, Ni)-ZnO nanoparticles in the plating bath increased up to 6 g/L, *R_ct_* reached the maximum value of 8.7 kΩ ∙ cm^2^, and the corrosion resistance of the coating was the best. As the addition of (Cu, Ni)-ZnO nanoparticles continued to increase, the *R_ct_* values of the Cu-Ni-ZnO nanocomposite coatings decreased, resulting in a decrease in corrosion resistance, which was consistent with the polarization results shown in Table 2. Supersaturated (Cu, Ni)-ZnO nanoparticles are prone to agglomeration and are not easy to deposit on the surface of the coatings. Moreover, the (Cu, Ni)-ZnO nanoparticles deposited on the coatings surface became very loose due to agglomeration, thus reducing the corrosion resistance of the nanocomposite coatings.

In general, the improvement in the corrosion resistance of nanocomposite coatings by nanoparticles was mainly related to the anti-corrosive properties of nanoparticles, defects inside the coatings such as micro-voids and micro-cracks, and the dispersion of nanoparticles in the coatings [45,51]. Combined with the results in Figure 2 and Figure 4, when the addition of (Cu, Ni)-ZnO in the electroplating solution increased to 6 g/L, the concentration of nanoparticles in the coatings reached the peak. At this time, the microstructure of the coatings became finer, more uniform, and denser, which was beneficial for the improvement in corrosion resistance. At the same time, the addition of (Cu, Ni)-ZnO nanoparticles filled the cracks and holes in the deposition of the Cu-Ni alloy, hindered the generation of defect corrosion, and further improved the corrosion resistance of the coatings [43,52]. However, along with increasing the addition of (Cu, Ni)-ZnO nanoparticles in the electroplating plating bath, the concentration of nanoparticles in the nanocomposite coatings decreased, which increased the crystallite size of the coatings and eventually led to a decrease in corrosion resistance.

### 3.5. Photocatalytic Performance

The lattice defects caused by Cu and Ni co-doping enhanced the absorption of photons, thereby broadening the light absorption efficiency of ZnO nanoparticles under visible light [36]. The optical band gaps of (Cu, Ni)-ZnO nanoparticles were evaluated by the Tauc relation. Compared with the undoped ZnO, the *E*_g_ (optical band gap energies) value of 2% Cu + 2% Ni co-doped ZnO nanoparticles ranged from 3.083 to 2.887 eV [36]. The reduced *E_g_* made the (Cu, Ni)-ZnO nanoparticles exhibit a better photocatalytic performance.

An ultraviolet visible spectrophotometer was used to test the photocatalytic performance of the Cu-Ni-ZnO nanocomposite coatings. The prepared nanocomposite coatings were put into 100 mL of 8 mg/L RhB solution and irradiated with ultraviolet light. The photocatalytic performance of the Cu-Ni-ZnO nanocomposite coatings was analyzed according to the change in the absorbance value of the RhB solution. The absorbance values of the Cu-Ni-ZnO coatings in RhB solution after UV irradiation for 0–5 h are listed in Table 4. The results show that the absorbance values of the RhB solution decreased in varying degrees with the prolongation of the UV irradiation time for all of the coatings. Compared with the Cu-Ni alloy coating, the absorbance values of the Cu-Ni-ZnO nanocomposite coatings decreased more significantly. With the increase in (Cu, Ni)-ZnO nanoparticles addition, the absorbance values of the nanocomposite coatings decreased gradually. The absorbance values of the RhB solution of the Cu-Ni-ZnO nanocomposite coatings with (Cu, Ni)-ZnO addition of 6 g/L decreased the most after 5 h of UV irradiation, exhibiting the best photocatalytic degradation performance.

According to the absorbance values in Table 4 and the decolorization rate formula, the relationship between the decolorization rate and UV irradiation time of the Cu-Ni-ZnO nanocomposite coatings were obtained, as shown in Figure 10. The decolorization rate *η* of all of the samples showed an upward trend with the increase in illumination time. It can be seen that the *η* of the Cu-Ni alloy coating changed very little after 5 h of ultraviolet light irradiation. The *η* of the RhB solution for Cu-Ni alloy coating was only 8.84%, indicating that the degradation effect of the Cu-Ni alloy coating on RhB was poor, and the change in their *η* was caused by adsorption. However, the addition of (Cu, Ni)-ZnO nanoparticles significantly improved the *η* of the Cu-Ni-ZnO nanocomposite coatings.

When the addition of (Cu, Ni)-ZnO nanoparticles in the plating solution was 6 g/L, the *η* of RhB solution reached the highest percentage of 28.73% after 5 h of UV irradiation. At this concentration, the Cu-Ni-ZnO nanocomposite coating exhibited the best photocatalytic effect, which was mainly because more nanoparticles were compounded during the co-deposition process. When the addition of (Cu, Ni)-ZnO nanoparticles was 8 g/L, the *η* of Cu-Ni-ZnO nanocomposite coating to RhB solution decreased, which was due to the reduction in (Cu, Ni)-ZnO nanoparticles in the coating and possible agglomeration, thus reducing the photocatalytic performance. Although (Cu, Ni)-ZnO nanoparticles had a higher photocatalytic effect, they were difficult to recycle and reuse. Electrodeposition of (Cu, Ni)-ZnO nanoparticles on the surface of the metal alloy by nanocomposite electroplating coud effectively improve the utilization rate and solve the problem of difficult powder recovery.

## 4. Conclusions

In this work, Cu and Ni co-doped ZnO nanoparticles were synthesized by the hydrothermal method, and then nanocomposite coatings with different (Cu, Ni)-ZnO additions were prepared using nanocomposite plating technology. The effects of nanoparticle addition on the phase structure, surface morphology, thickness, microhardness, corrosion resistance, and photocatalytic performance of the coatings were studied.

(1)Cu-Ni-ZnO nanocomposite coatings had diffraction peaks on (111), (200), and (220) crystal planes with a wurtzite structure. The surface morphology of the nanocomposite coatings was cauliflower-like, being more uniform and dense. The microhardness, corrosion resistance, and photocatalytic performance of the nanocomposite coatings were obviously superior to those of the Cu-Ni alloy coating.(2)The addition of (Cu, Ni)-ZnO improved the performance of the nanocomposite coatings as a whole, and the various properties reached the peak at 6 g/L. At this concentration, the minimum crystallite size was 15.5 nm and the microhardness was 658 HV. The corrosion resistance of the coatings was the best with the minimum corrosion current density of 2.36 × 10^−6^ A/cm^2^ and the maximum *R_ct_* of 8.7 kΩ ∙ cm^2^. Meanwhile, the decolorization rate of the RhB solution reached the highest rate of 28.73% after 5 h of UV irradiation.(3)With the increase in (Cu, Ni)-ZnO additions, the concentration of nanoparticles in the nanocomposite coatings increased gradually. The increased concentration of nanoparticles in the coatings favored a finer, more uniform and denser microstructure, which can further improve the corrosion resistance and photocatalytic degradation performance of the coatings. However, further increment in the concentration of (Cu, Ni)-ZnO nanoparticles in the plating bath resulted in an overall performance decrement, which was due to the reduction in the nanoparticles in the coatings and possible agglomeration.

## Figures and Tables

**Figure 1 materials-16-02746-f001:**
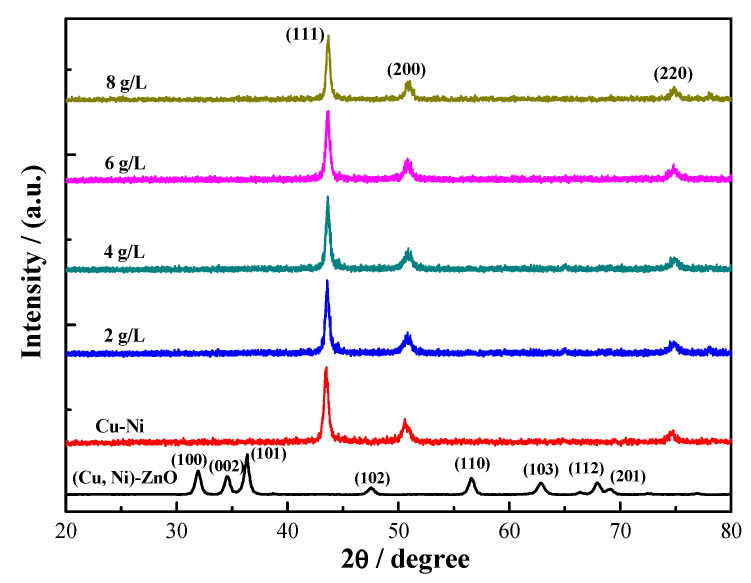
XRD patterns of the (Cu, Ni)-ZnO nanopowder and Cu-Ni-ZnO nanocomposite coatings.

**Figure 2 materials-16-02746-f002:**
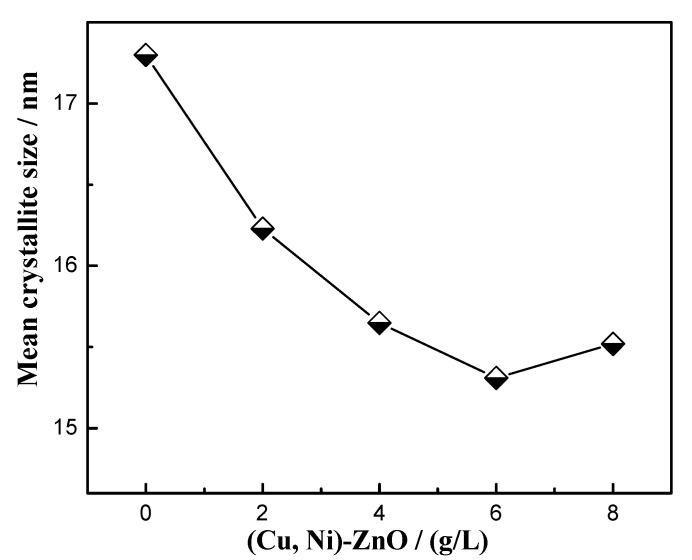
The mean crystallite size of Cu-Ni-ZnO nanocomposite coatings with different (Cu, Ni)-ZnO additions.

**Figure 3 materials-16-02746-f003:**
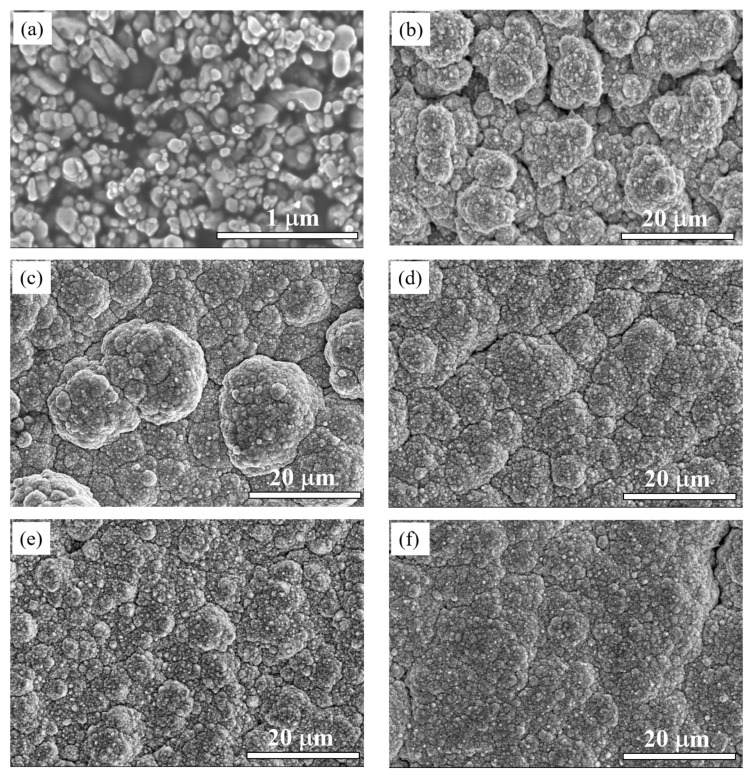
SEM images of Cu-Ni-ZnO nanocomposite coatings with different (Cu, Ni)-ZnO additions: (**a**) (Cu, Ni)-ZnO nanoparticles, (**b**) 0 g/L, (**c**) 2 g/L, (**d**) 4 g/L, (**e**) 6 g/L, and (**f**) 8 g/L.

**Figure 4 materials-16-02746-f004:**
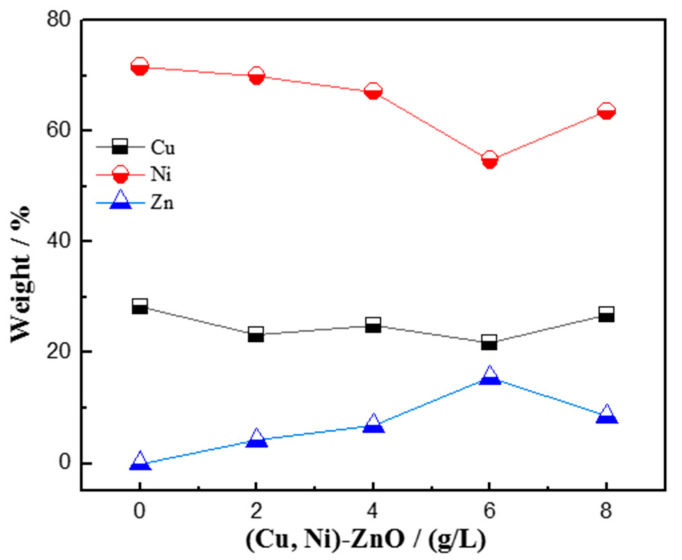
Composition of Cu, Ni, and Zn in Cu-Ni-ZnO nanocomposite coatings with different (Cu, Ni)-ZnO additions.

**Figure 5 materials-16-02746-f005:**
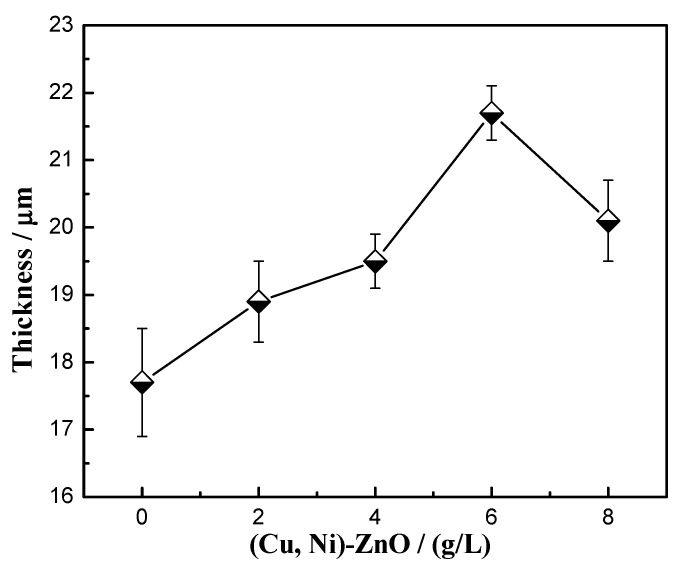
Thickness of Cu-Ni-ZnO nanocomposite coatings with different (Cu, Ni)-ZnO additions.

**Figure 6 materials-16-02746-f006:**
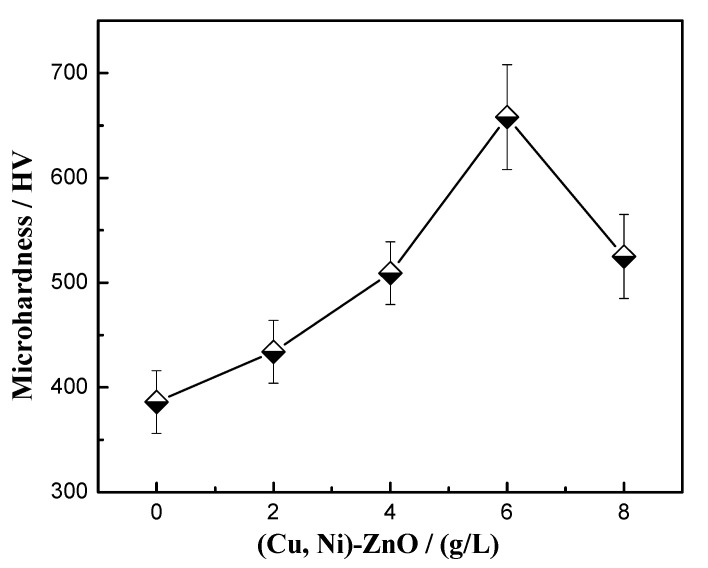
Microhardness of Cu-Ni-ZnO nanocomposite coatings with different (Cu, Ni)-ZnO additions.

**Figure 7 materials-16-02746-f007:**
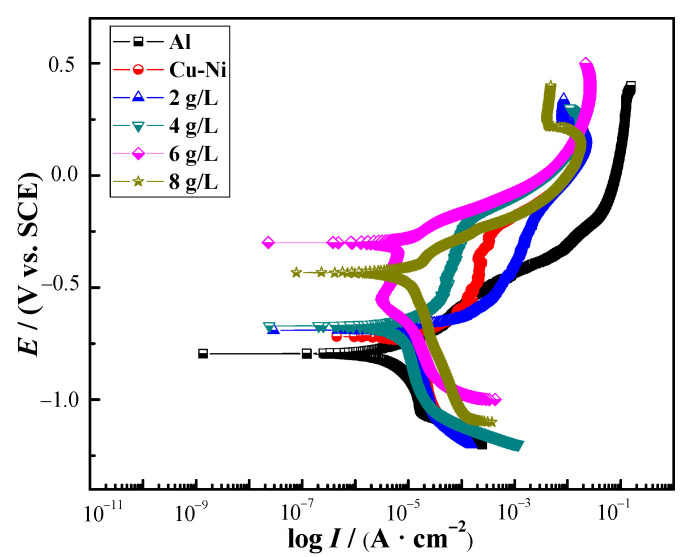
Polarization curves of Cu-Ni-ZnO nanocomposites with different (Cu, Ni)-ZnO additions.

**Figure 8 materials-16-02746-f008:**
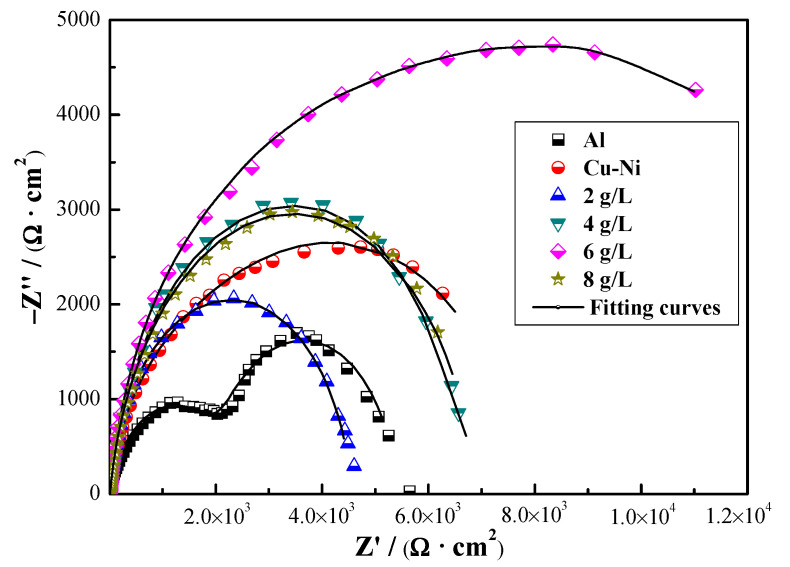
Nyquist diagrams of Cu-Ni-ZnO nanocomposite coatings with different (Cu, Ni)-ZnO additions.

**Figure 9 materials-16-02746-f009:**
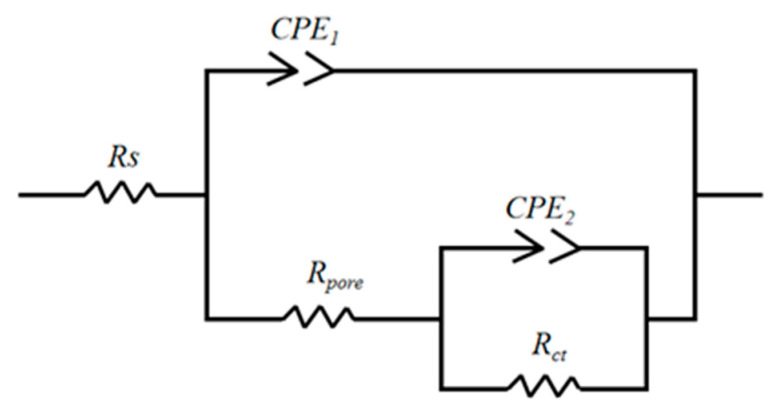
Equivalent circuit for fitting the EIS of the Cu-Ni-ZnO nanocomposite coatings.

**Figure 10 materials-16-02746-f010:**
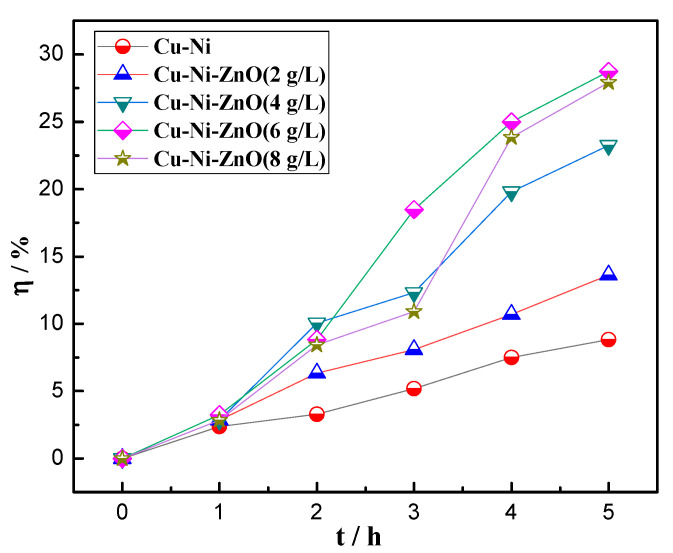
Illumination time and decolorization rate of Cu-Ni-ZnO nanocomposite coatings with different (Cu, Ni)-ZnO additions.

**Table 1 materials-16-02746-t001:** The chemical composition and process parameters of electroplating solution for preparing nanocomposite coatings.

Composition	Concentration	Parameters	Range
CuSO_4_·5H_2_O	20 g/L	Temperature	45 °C
NiSO_4_·6H_2_O	85 g/L	Current density	25 mA·cm^−2^
C6H_5_O_7_Na_3_·2H_2_O	75 g/L	Deposition time	45 min
C_12_H_25_SO_4_Na	0.2 g/L	pH	7
(Cu, Ni)-ZnO	2–8 g/L	stirring rate	300 rpm

**Table 2 materials-16-02746-t002:** Tafel fitting results of polarization curves of Cu-Ni-ZnO nanocomposite coatings with different (Cu, Ni)-ZnO additions.

Concentration/(g/L)	*I_corr_*/(A ∙ cm^−2^)	*E_corr_*/V	*E_pit_*/V
Substrate	2.73 × 10^−6^	−0.779	−0.466
0	1.29 × 10^−5^	−0.677	−0.222
2	7.41 × 10^−6^	−0.664	−0.183
4	7.22 × 10^−6^	−0.658	−0.178
6	2.36 × 10^−6^	−0.301	−0.098
8	8.75 × 10^−6^	−0.421	−0.177

**Table 3 materials-16-02746-t003:** Fitting results of the equivalent circuit of Cu-Ni-ZnO nanocomposite coatings with different (Cu, Ni)-ZnO additions.

Concentration /(g/L)	*R_s_* /(Ω·cm^2^)	*Q_1_* /(S·cm^−2^·s^−*n*^)	*n*	*R_pore_* /(kΩ·cm^2^)	*Q_2_* /(S·cm^−2^·s^−*n*^)	*n*	*R_ct_* /(kΩ·cm^2^)
Al substrate	10.45	9.96 × 10^−5^	0.84	2.36	9.29 × 10^−4^	0.98	2.89
0	12.26	5.79 × 10^−5^	0.91	3.63	2.58 × 10^−6^	0.60	4.42
2	8.99	8.93 × 10^−5^	0.94	4.91	4.52 × 10^−4^	0.93	2.53
4	15.77	6.80 × 10^−4^	0.92	2.91	4.18 × 10^−4^	0.90	6.51
6	11.45	6.58 × 10^−4^	0.91	4.43	1.13 × 10^−5^	1	8.70
8	13.25	6.58 × 10^−4^	0.90	3.97	1.24 × 10^−4^	0.98	6.26

**Table 4 materials-16-02746-t004:** Absorbance values of Cu-Ni-ZnO nanocomposite coatings with different (Cu, Ni)-ZnO addition under ultraviolet light irradiation.

Radiation time/h	0	1	2	3	4	5
Cu-Ni	0.51705	0.50481	0.50005	0.49029	0.47829	0.47136
2 g/L	0.51705	0.50228	0.48429	0.47530	0.45872	0.44662
4 g/L	0.51705	0.50235	0.46503	0.45332	0.41467	0.39682
6 g/L	0.51705	0.50028	0.47136	0.42156	0.38779	0.36850
8 g/L	0.51705	0.50235	0.47348	0.46065	0.39378	0.37269

## Data Availability

Not applicable.

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
