# Peer review of "Preparation and Properties of (Cu, Ni) Co-Doped ZnO Nanoparticle-Reinforced Cu-Ni Nanocomposite Coatings"

_materials, 2023, doi:10.3390/ma16072746_

Round 1

Reviewer 1 Report

The authors have discussed the effect of Cu-Ni-ZnO concentration to improve the nano-composite coating properties. The best corrosion resistance and photo-catalytic properties are found at 6 g/L concentrations. The experimentation is adequate, flow is nice, but a few explanations are missing. The following comments are provided to improve the quality of the manuscript:

1. Can authors add a novelty statement in the introduction section?

2. Lines 99, 102: the valency of the elements should be in subscript.

3. Figure 1: Include ZnO diffraction data to have a better understanding of peak collapse after Cu-Ni inclusions.

4. Figure 2: Why the grain size slightly increases at 8 g/L concentration?

5. Line 236 - 237: the meaning of the sentence is not clear. do authors try to mean that the load at the indenter hinders the dislocation movement? how is that possible?

6. Figure 5: error bars should be included in the thickness measurement.

7. The SEM images show least presence of defects, voids and cracks for 8 g/L concentration sample, but the corrosion resistance is quite high for this. Can authors provide an explanation for this?

8. The 'section 4: discussion' is from the MDPI template and that should be deleted from the text.

Reviewer 2 Report

Dear Authority,

The manuscript entitled ‘Preparation and properties of (Cu, Ni) co-doped ZnO nanoparticles reinforced Cu-Ni nanocomposite coatingsinvestigates the effect of Cu-Ni doped ZnO addition to Cu-Ni coatings in terms of coating morphology, corrosion resistance, and photocatalytic properties. The increment in nanoparticle content on the coating leads improvement on hardness and corrosion resistance. But, when the nanoparticle amount reaches 8 g/L, hardness values is deteriorated. The grain morphology is detected in nanoscale range (15.31 nm). Addition of nanoparticles also helps to enhance density of coating with smaller grain morphology which eventually determine corrosion resistance and photocatalytic degradation performance of coatings.   

I think, the paper includes important information and data which will be useful for literature. It could be considered for publication after minor correction according to following comments/recommendations;

1- Nanoparticle composition is defined as 2% Cu + 2% Ni co-doped ZnO. What is the composition unit? (wt%, vol%, or at%) please state in manuscript body.

2- I think, the crystallite size measurement Debye-Scherrer formula is not reliable, because both Cu and Ni have peaks at same peak position. For instance major peak of Cu and Ni is around θ=43.5o, so two phases make contribution the same peak. To make realible calculation peaks belongs to two phases needs to be split. Otherwise, the peak breadth becomes wider and crystallite size is calculated smaller.

3- The discussion and conclusion are numbered as 4. Please correct them.

The manuscript can be published in Materials after these minor corrections.

Best wishes,

Author Response

   On behalf of my co-authors, thank you for your letter and for the reviewers’ comments concerning our manuscript entitled “Preparation and properties of (Cu, Ni) co-doped ZnO nanoparticles reinforced Cu-Ni nanocomposite coatings”. Those comments are all valuable and very helpful for revising and improving our paper, as well as the important guiding significance to our researches. We have studied comments carefully and have made correction which we hope meet with approval.

 We tried our best to improve the manuscript and made some changes in the manuscript.  These changes will not influence the content and framework of the paper. And here we did not list the changes but marked in red in revised paper.

   We appreciate Reviewers’ warm work earnestly, and hope that the correction will meet with approval.

  Once again, thank you very much for your comments and suggestions.

  Thank you and best regards.

Reviewer 3 Report

The authors studied the effect of Cu and Ni codoped ZnO powders on the physico-chemical properties, mechanical properties, photocatalytic properties, and corrosion resistance of Cu-Ni coating.

This work has some merit, however it should be improved following these comments:

1/ In the first part of the introduction the authors put different ZnO applications, but they should add storage application and they can use these new refences (Enhancing the electrical conductivity and the dielectric features of ZnO nanoparticles through Co doping effect for energy storage applications. DOI: 10.1007/s10854-022-09470-5) and (Enhancing the electrical and dielectric properties of ZnO nanoparticles through Fe doping for electric storage applications. DOI: 10.1007/s10854-020-04923-1).

2/ Why the authors choose aluminium cathode and not copper?

3/ The authors should describe how they obtained the different parameters in table 1. Why the authors used a neutral electroplating solution and not acid?

4/ Why the authors choose 2% of Cu and 2% of Ni as dopant in ZnO?

5/ The authors should introduce the XRD pattern of Cu/Ni codoped ZnO in figure 1. Why the characteristic peaks of ZnO are not present in the different patterns of nanocomposites?

6/ In line 158, the authors speak about grain size from XRD measurements, but here the crystallite size is deduced and not grain size using Debye-Scherrer method.

7/ In figure 8, the authors should add the fitting curves to obtain the fitting results of equivalent circuit.

8/ The authors should respect the same curve colour for each sample in different part of the manuscript.

9/ More details should be added in the discussion part to correlate between different results.

Author Response

Dear Reviewers:

   On behalf of my co-authors, thank you for your letter and for the reviewers’ comments concerning our manuscript entitled “Preparation and properties of (Cu, Ni) co-doped ZnO nanoparticles reinforced Cu-Ni nanocomposite coatings”. Those comments are all valuable and very helpful for revising and improving our paper, as well as the important guiding significance to our researches. We have studied comments carefully and have made correction which we hope meet with approval.

   We tried our best to improve the manuscript and made some changes in the manuscript.  These changes will not influence the content and framework of the paper. And here we did not list the changes but marked in red in revised paper.

   We appreciate for Reviewers’ warm work earnestly, and hope that the correction will meet with approval.

   Once again, thank you very much for your comments and suggestions.

   Thank you and best regards.

Round 2

Reviewer 3 Report

The authors corrected and improved the manuscript as recommended by the reviewer and they answered satisfacory to the questions. 

The reviewer recommends the publication of this work in Materials.

Author Response

Dear Reviewer:

   Thanks very much for your good comments and consideration on publication of our paper. On behalf of my co-authors, we would like to express our great appreciation to you.

  Thank you and best regards.